# Effect of Varying the Ratio of Carbon Black to Vapor-Grown Carbon Fibers in the Separator on the Performance of Li–S Batteries

**DOI:** 10.3390/nano9030436

**Published:** 2019-03-15

**Authors:** Hearin Jo, Jeonghun Oh, Yong Min Lee, Myung-Hyun Ryou

**Affiliations:** 1Department of Chemical and Biological Engineering, Hanbat National University, 125 Dongseo-daero, Yuseong-gu, Daejeon 34158, Korea; hearin.jo124@gmail.com (H.J.); jeonghun.ohh@gmail.com (J.O.); 2Department of Energy Systems Engineering, Daegu Gyeongbuk Institute of Science and Technology (DGIST), 333 Techno Jungang-Daero, Daegu 42988, Korea; yongmin.lee@dgist.ac.kr

**Keywords:** carbon-coated separator, polysulfide, shuttle effect, lithium–sulfur batteries

## Abstract

Lithium–sulfur (Li–S) batteries are expected to be very useful for next-generation transportation and grid storage because of their high energy density and low cost. However, their low active material utilization and poor cycle life limit their practical application. The use of a carbon-coated separator in these batteries serves to inhibit the migration of the lithium polysulfide intermediate and increases the recyclability. We report the extent to which the electrochemical performance of Li–S battery systems depends on the characteristics of the carbon coating of the separator. Carbon-coated separators containing different ratios of carbon black (Super-P) and vapor-grown carbon fibers (VGCFs) were prepared and evaluated in Li–S batteries. The results showed that larger amounts of Super-P on the carbon-coated separator enhanced the electrochemical performance of Li–S batteries; for instance, the pure Super-P coating exhibited the highest discharge capacity (602.1 mAh g^−1^ at 150 cycles) with a Coulombic efficiency exceeding 95%. Furthermore, the separators with the pure Super-P coating had a smaller pore structure, and hence, limited polysulfide migration, compared to separators containing Super-P/VGCF mixtures. These results indicate that it is necessary to control the porosity of the porous membrane to control the movement of the lithium polysulfide.

## 1. Introduction

The continuously increasing worldwide demand for energy has resulted in energy storage systems becoming essential for the successful implementation of various electric devices such as electric vehicles, portable electronic devices, and energy storage systems [1,2,3,4,5,6]. Lithium–sulfur batteries have been regarded as promising candidates due to their high theoretical capacity (1675 mA∙h∙g^−1^), low cost, and the environmentally friendly characteristics of sulfur. Despite the many advantages of sulfur cathodes, Li–S batteries have poor cycle performance due to following chronic drawbacks, which lead failure at successful commercialization [7,8,9,10,11]: (1) Sulfur has insulating properties that interfere with uniform electrons throughout the active materials during operation. As a result, poor utilization of the active sulfur material occurs during electrochemical reactions. (2) Lithium polysulfides (Li_2_S_x_, 4 ≤ x ≤ 8), the intermediates of sulfur during the charging/discharging processes, are a fatal component impeding the cycle performance of Li–S batteries. Lithium polysulfide dissolves easily in the electrolyte from the sulfur cathode and consumes electrons directly from the both electrodes, cathodes, and anodes inside the battery system, unlike conventional battery systems, where electrons are consumed along the conductors. This series of internal cyclic electron consumption is called the “shutting effect” [12,13,14].

Various approaches have been proposed to overcome the drawbacks of the sulfur cathodes described above to improve Li–S batteries through the modification of the sulfur cathodes and electrolyte systems, and in this regard, advanced sulfur/carbon composites [3,15,16], functional polymeric binders [17,18], sulfur/metal–organic frameworks [19,20], solid electrolytes [21,22,23], and functional additives [24] have been studied.

Although the separators are one of the main components of the battery system (composed of anodes, cathodes, electrolytes, and separators), the significance of the separators for cycle performance of Li–S batteries was underestimated until a functional separator containing a carbon coating layer was proposed [25,26,27,28]. A carbon coating layer not only provides an electron path to the sulfur cathodes but also effectively reduces the migration of lithium polysulfides through the separators and leaves them on the sulfur cathode surface. As a result, the carbon-coated separators improve the cycle performance of the Li–S batteries by helping the sulfur cathodes to reuse lithium polysulfides during the repeated charging/discharging processes. Furthermore, from a practical point of view, the use of functional separators is not only economical but is also advantageous over other existing approaches associated with sulfur cathode and electrolyte modification, since it can be applied to a variety of existing technologies.

We have focused on the selection guidelines for carbon materials for Li−S batteries, and to do so we fabricated carbon-coated separators using two representative commercial carbon materials, carbon black (Super-P) and vapor-grown carbon fibers (VGCFs). We see that when Super-P and VGCFs are used as conductive additives for lithium–cobalt–oxide (LiCoO_2_) cathodes, there is a synergy that leads to improved LiCoO_2_ performance in unexpected combinations. Taking this into account, it is thought that the combination of Super-P and VGCFs for carbon-coated separator needs to be investigated for Li−S batteries, because changing the ratio of Super-P and VGCFs significantly changes the electrical conductivity and porosity of the coating separators.

The design of carbon-coated separators for Li–S batteries has to carefully consider the porosity of the coating layer. For instance, the use of carbon-coated separators which are not porous with a dense surface structure would prevent the liquid electrolytes containing the polysulfide from flowing through the layer fluently, thereby making it difficult for the polysulfide to migrate to the Li metal surface. From a kinetic point of view, however, limited movement of the liquid electrolyte inside the batteries may result in electrochemical performance degradation due to low reaction rates. On the other hand, highly porous, carbon-coated separators would be beneficial to the kinetic behavior of the Li–S battery system but would be vulnerable to polysulfide migration. In this study, the porosity of the carbon-coated separator was adjusted by using mixtures of different types of carbon materials, VGCFs and Super-P. Simply changing the VGCF to Super-P ratio in the carbon-coated separator enabled us to optimize the porosity of the separator as an effective approach to obtain a stable high-performance Li–S battery with exceptional rate capability.

## 2. Materials and Methods

### 2.1. Materials

Sulfur (100 mesh, Sigma–Aldrich, St. Louis, MO, USA), Ketjenblack (Ketjenblack^®^EC-600JD, AkzoNobel, Amsterdam, Netherlands), poly(vinylidene fluoride) (PVdF, KF-1300, Kureha, Iwaki, Japan, *M*_w_ = 350000), vapor-grown carbon fibers (VGCFs, Showa Denko K.K, Tokyo, Japan), Super-P (Li-conductive) (IMERYS, Paris, France), poly(vinylidene fluoride-co-hexafluoropropylene) (PVdF-HFP, Kynar Flex^®^ 2801, Arkema Inc., Colombes, France), *N*-methyl-2-pyrrolidone (NMP, Sigma–Aldrich), 1,3-dioxolane (DOL, Sigma–Aldrich, St. Louis, MO, USA), 1,2-dimethoxyethane (DME, Sigma–Aldrich, St. Louis, MO, USA), LiTFSI (Enchem, Jecheon, Korea), Li metal foil (thickness = 200 µm, Honjo Metal Co., Osaka, Japan), and polypropylene (PP) separators (thickness = 25 μm, Celgard 2400, Celgard^®^, Charlotte, NC, USA) were used as separators.

### 2.2. Preparation of the VGCF and Super-P Carbon Composite

The VGCF and Super-P powder, both of which were purchased from commercial corporations, were first mixed together in a specific weight ratio and then dispersed in a poly(vinylidene fluoride-co-hexafluoropropylene) (PVdF-HFP, Kynar Flex^®^ 2801, Arkema Inc., Colombes, France) solution (2 wt.% of PVdF-HFP dissolved in *N*-methyl-2-pyrrolidone (NMP, Sigma–Aldrich, St. Louis, MO, USA)) to form a uniform slurry.

### 2.3. Preparation of the Modified Separators

In this work, the separators were modified by fabricating them with three different coatings by using slurries with different compositions. The first (VGCF-coated separator) consisted of 70 wt.% VGCF and 30 wt.% PVdF-HFP; The second (VGCF and Super-P composite-coated separator) consisted of 35 wt.% VGCF, 35 wt.% Super-P, and 30 wt.% PVdF-HFP; the third (Super-P-coated separator) consisted of 70 wt.% Super-P and 30 wt.% PVdF-HFP. These three kinds of separators were prepared by modifying the conventional separator (PP separator, Celgard 2400) by directly applying a coating of the three slurries mentioned above on the PP separator using a gap-controlled doctor blade. After they were dried at 50 °C for 12 h in the oven, the three kinds of modified separators were punched into circular disks with a diameter of 18 mm. The fabricated layers of carbon coating had an average thickness of ~10 µm, which is the minimum value to ensure reasonable polysulfide inhibition behavior. The coating thickness issue will be further discussed in polysulfide diffusion experiments corresponding to Figure 2. The carbon loadings of the VGCF-coated separator, VGCF and Super-P composite-coated separator, and Super-P-coated separator were 0.18, 0.20, and 0.33 mg cm^−2^, respectively.

### 2.4. Preparation of the Sulfur Cathode and Cell Assembly

A sulfur/carbon (S/C) (Ketjen black) composite (S/C = 80/20 in weight) was prepared with the melt diffusion method. A slurry consisting of the S/C composite (70 wt.%), vapor grown carbon fiber (20 wt.%), and PVdF (10 wt.%) as a binder, was poured onto aluminum foil. Then the coated foil was dried at 50 °C for 12 h. Finally, the sulfur cathode was roll-pressed and punched into circular disks with a diameter of 12 mm. The areal loading of sulfur for the as-prepared electrodes ranged from 1.3 mg∙cm^−2^ to 1.5 mg∙cm^−2^. The electrochemical properties were tested by assembling 2032 coin-type half-cells using the sulfur electrodes, VGCF/Super-P coated separators, and Li metal as the counter electrode. The electrolyte was 1 M LiTFSI (lithium bis (trifluoromethanesulfonyl) imide) in DOL and DME (1:1 by volume) with 0.2 M LiNO_3_ as an additive. To standardize the measurement protocol, the amount of electrolyte added to each cell was controlled to 200 µL. Cell assembly was carried out in an argon-filled glove box, and all capacity values were calculated based on the sulfur mass.

### 2.5. Electrochemical Testing

After assembly, the coin cells (sulfur/Li metal) were stored for 12 h before the electrochemical measurements. Cycle performance was evaluated by cycling the unit cells over different potential ranges (1.9–2.8 V versus Li/Li^+^) in a constant current (CC) mode during both the charging and discharging processes at a constant current density C/2 (resp. 0.76 mA∙cm^−2^ for sulfur) using a charge/discharge cycler (PNE Solution, Suwon, Korea) at 25 °C. The cycle performance was evaluated at 1 C (CC during the charge and discharge processes within the same potential ranges.) The rate capability was evaluated by increasing the discharge current densities from C/5 to 3 C (C/5, C/2, 1 C, 2 C, and 3 C). The cells were discharged in CC mode while maintaining a charging current density of C/2 in CC mode.

### 2.6. Characterization and Electrochemical Measurements

After the electrochemical investigations were performed, the fully charged cells up to 2.8 V versus Li/Li^+^ were carefully disassembled in a dry Ar-filled glove box. The samples were washed several times with dimethyl carbonate (DMC, anhydrous, >99%, S Sigma–Aldrich, St. Louis, MO, USA), and then dried overnight under vacuum before observation. The samples were analyzed by performing field emission scanning electron microscopy with energy-dispersive X-ray analysis (FE-SEM/EDX, S-4800, Hitachi, Tokyo, Japan).

The Gurley number was evaluated using a densometer (4110N, Thwing-Albert, West Berlin, NJ USA) by measuring the time required for passing 100 mL of air through separators under 6.52 psi pressure [29].

## 3. Results and Discussion

### 3.1. Morphology and Physical Characterization of the Carbon-Coated Separator

For simplicity, the carbon-coated separator containing pure Super-P, the Super-P/VGCF combination, and pure VGCF are denoted as the Super-P, Super-P/VGCF, and VGCF separators, respectively.

Figure 1 shows the surface morphologies of each of these carbon-coated separators. The Super-P separator had a dense surface structure in powder form with an average particle size of ~40 nm. The larger size columnar VGCF particles (average particle diameter = ~150 nm, average length = 15 µm) resulted in the VGCF separator showing the most porous surface structure. Thus, the different dimensions and shapes of the Super-P and VGCF particles strongly influenced the surface morphology of the Super-P/VGCF separator, which was strongly dependent on the Super-P/VGCF ratio. The physical properties of each separator such as the Gurley number and surface resistance are listed in Table 1. All of the carbon-coated separators (the Super-P, Super-P/VGCF, and VGCF separators) had higher Gurley numbers than the bare uncoated separator (Celgard 2400). The additional carbon coating layer of ~9 μm with high tortuosity played the role of a gas barrier.

On the other hand, VGCF exhibited the lowest surface resistance, which is in good agreement with our previous study [30]. Similar to the present study, the previous study investigated the effect of various types of conductive additives (pure Super-P, pure VGCF, and a mixture of Super-P and VGCF) on lithium–cobalt–oxide (LiCoO_2_) cathodes. The LiCoO_2_ cathodes containing pure VGCF revealed the lowest surface resistance because VGCF builds an “expressway” for electron transfer, which facilitates electron transfer across the cathode.

### 3.2. Polysulfide Suppression Behaviors of Carbon-Coated Separators

In general, separators for batteries are placed between the cathode and anode and are designed to have a highly porous structure to allow ion migration through the pores. Separators with high porosity containing more massive amounts of liquid electrolyte result in improved ion mobility [31,32]. However, a Li–S battery with highly porous separators can increase the mobility of polysulfides, resulting in more severe polysulfide shuttle behavior which can critically affect the cycle performance [33]. Taking this into account, the polysulfide inhibition behavior of carbon-coated separators prepared with various Super-P and VGCF ratios was investigated.

As shown in Figure 2, the polysulfide permeability of the separator was examined. The inner glass tube was filled with a mixture consisting of 15 mL of a solution of 0.43 M Li_2_S_6_ and 15 mL of DME/DOL (1:1, by vol.). The outer glass tube was filled with 30 mL of DME/DOL (1:1, by vol.). Because of the difference in polysulfide concentration in the two tubes, the polysulfide spread to the outer glass tube over time. The bare separator (Celgard 2400) showed the fastest polysulfide diffusion, whereas the Super-P separator exhibited the best protection against polysulfide diffusion. After 6 h, the solution in the outer glass tube wrapped with the bare separator turned brown, but the Super-P separator ensured that the solution in the outer glass tube remained transparent even after 12 h. On the other hand, when the carbon coating layer was less than 10 μm, the diffusion of the polysulfide could not be adequately suppressed even by the Super-P separator. Thus, an increase in the Super-P ratio more effectively inhibited the diffusion of polysulfide. These results indicate that Super-P effectively immobilizes polysulfide inside nano-sized porous structures.

### 3.3. Electrochemical Performance

Figure 3a shows the galvanostatic discharge/charge potential profiles during pre-cycling for the Li–S battery containing the three types of carbon-coated separators (Super-P, Super-P/VGCF (5:5 by wt.%), and VGCF) measured at C/5. During the discharging process (lithiation), the upper discharge plateau near 2.4 V represents the conversion of sulfur (S_8_) to soluble polysulfide (Li_2_S_x_, 4 ≤ x ≤ 8), and the lower discharge plateau near 2.1 V represents the conversion of soluble polysulfide (Li_2_S_x_, 4 ≤ x ≤ 8) to solid polysulfide (Li_2_S_2_/Li_2_S) [34]. During the charging process, the first long and flat plateau near 2.2 V corresponds to the conversion of solid polysulfide (Li_2_S_2_/Li_2_S) to soluble polysulfide (Li_2_S_x_, 4 ≤ x ≤ 8), and the plateau near 2.35 V corresponds to the conversion of soluble polysulfide (Li_2_S_x_, 4 ≤ x ≤ 8) to sulfur (S_8_) [34]. Although the theoretical potentials of each plateau are 2.18 and 2.33 V versus Li/Li^+^, the plateaus generally differed during charging and discharging because of the IR drop ascribed to the high internal resistance of Li–S batteries [35]. The Li–S batteries containing Super-P exhibited the highest initial discharge capacity (1219.5 mA∙h∙g^−1^) with the highest Coulombic efficiency (100%). The first discharge capacity of each cell containing the carbon-coated separator exceeded that of the bare separator (Super-P = 1213.0 mA∙h∙g^−1^, Super-P/VGCF = 1158.4 mA∙h∙g^−1^, VGCF = 1120.7 mA∙h∙g^−1^, bare PP = 1017.4 mA∙h∙g^−1^). The hysteresis shown in Figure 3a usually can be observed from the initial discharge of other conversion electrode materials because this is attributed to the poor electrical contact of the initial grain boundaries between active materials and conducting carbon materials [36]. As can be seen in Figure 3b, the hysteresis observed under 2.0 V versus Li/Li^+^ during pre-cycling (associated with Figure 3a) disappeared. Considering this, we can infer that the Li−S batteries were stabilized during pre-cycling.

As shown in Figure 3c, the cycle performance of the Li–S batteries containing the carbon-coated separators was evaluated at a discharging rate of 1 C. The unit cells containing larger amounts of Super-P showed a higher initial discharge capacity (after the first cycle, Super-P = 984.3 mA∙h∙g^−1^, Super-P/VGCF = 903.0 mA∙h∙g^−1^, VGCF = 795.9 mA∙h∙g^−1^, bare PP = 690.4 mA∙h∙g^−1^). The cycle performance of Li–S batteries was greatly improved when larger amounts of Super-P were used (after 150 cycles, Super-P = 602.1 mA∙h∙g^−1^, Super-P/VGCF = 501.5 mA∙h∙g^−1^, VGCF = 301.0 mA∙h∙g^−1^, bare PP = 0 mA∙h∙g^−1^). Because the Gurley number is defined by passing a specific amount of air through the medium, the exact correlation between the polysulfide and the separators cannot be clearly explained. Nonetheless, it is plausible that separators with a high Gurley number are beneficial in inhibiting polysulfide migration since the Gurley number reflects the tortuosity of the separators [32,37]. With this in mind, the improved cycle performance of the Li–S unit cells containing Super-P separators is reasonable.

The rate capabilities of the Li–S batteries containing the various carbon-coated separators were also evaluated by increasing the discharging current density step-wise from C/5 (0.27 mA∙cm^−2^) to 3 C (4.02 mA∙cm^−2^) every seven cycles. As shown in Figure 3d, the rate capabilities of the Li–S unit cells were significantly improved when more substantial amounts of Super-P were used (at the 35th cycle for the 3 C rate: Super-P = 659.8 mA∙h∙g^−1^, Super-P/VGCF = 582.1 mA∙h∙g^−1^, VGCF = 509.5 mA∙h∙g^−1^, bare PP = 3.4 mA∙h∙g^−1^). These results were unusual because the Super-P separators with the highest Gurley number, and with the highest tortuosity, exhibited the highest rate capabilities. Given the results, we can infer that, of the two main factors, the tortuosity of the separators and migration of polysulfide, the latter is more decisive in determining the cycle performance as well as the rate capabilities of Li–S batteries.

### 3.4. Post-Mortem Analysis of Li–S Batteries after Cycling

After 20 cycles (corresponding to Figure 3c), the sulfur cathodes were retrieved from fully charged Li–S unit cells and the surface structures of the sulfur cathodes were observed using SEM. As shown in Figure 4, the morphological structure of the sulfur cathodes varied depending on the type of coating that was used on the separator. Deep holes were observed across the entire surface of the sulfur cathode in the Li–S unit cells containing bare separators (Figure 4a) and VGCF (Figure 4d). On the other hand, the sulfur cathodes of the Li–S unit cells containing larger amounts of Super-P showed a dense structure with fewer pores. Because polysulfide intermediates are highly soluble in electrolytes, these results are plausible because Super-P-rich separators more efficiently hinder the movement of polysulfides [28,38].

The surface structure of the cathode side of each separator was also observed using SEM and EDX. As shown in Figure 5, the morphological structures of the separators are almost similar to those shown in Figure 1. In contrast, the elemental composition determined by EDX showed that the Super-P-rich separator contained larger amounts of the element sulfur (Super-P = 6.54 wt.%, Super-P/VGCF = 2.00 wt.%, and VGCF = 1.08 wt.%). On the other hand, as shown in Figure 6, after exposure to the same experimental conditions, the Li metal surface was observed using SEM and EDX. Again, although the morphological structure was almost the same, the Li metal recovered from the disassembled Li–S unit cells containing Super-P-rich separators contained a smaller amount of the element sulfur on the surface (Super-P = 6.36 wt.%, Super-P/VGCF = 13.28 wt.% and VGCF = 16.14 wt.%). The sulfur element was originated from the sulfur-containing species, namely polysulfide and LiTFSI. The relationship between polysulfide and LiTFSI for electrochemical decomposition during discharge has not yet been clearly understood. Nonetheless, it can be easily deduced that the elemental change of the Li metal surface depends mainly on the amount of polysulfide. This is because the amount of LiTFSI is the same in all cases because the same amount of liquid electrolyte is used, but the amount of polysulfide changes during operation.

## 4. Conclusions

In this study, the effect of the Super-P/VGCF ratio of the carbon-coated separators on the electrochemical performance of Li–S batteries was investigated. Although the Super-P-rich separator exhibited the highest tortuosity with the highest Gurley number, Li–S unit cells containing the Super-P-rich separator showed superior cycle performance and rate capabilities compared to the other types of separators. This implies that polysulfide shuttling is the main factor determining the performance of Li–S batteries rather than the dynamic behavior of separators. Furthermore, we demonstrated that the migration of the soluble polysulfide was efficiently inhibited by the Super-P-rich separators, which prevented the polysulfide from reaching the Li metal surface (Figure 7). Consequently, manipulating the porosity of the porous membrane to control the migration of soluble lithium polysulfide is of key importance for the development of Li–S batteries.

## Figures and Tables

**Figure 1 nanomaterials-09-00436-f001:**
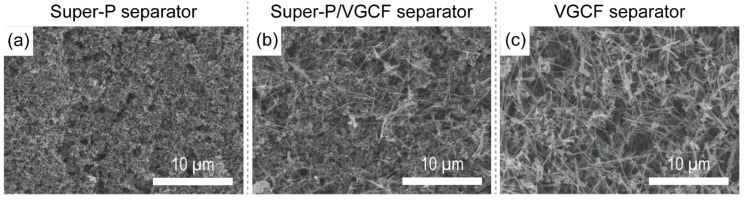
Scanning electron microscopy (SEM) images of the separator surface containing (**a**) Super-P; (**b**) Super-P/vapor-grown carbon fiber (VGCF) (5:5 by wt.%); and (**c**) VGCF.

**Figure 2 nanomaterials-09-00436-f002:**
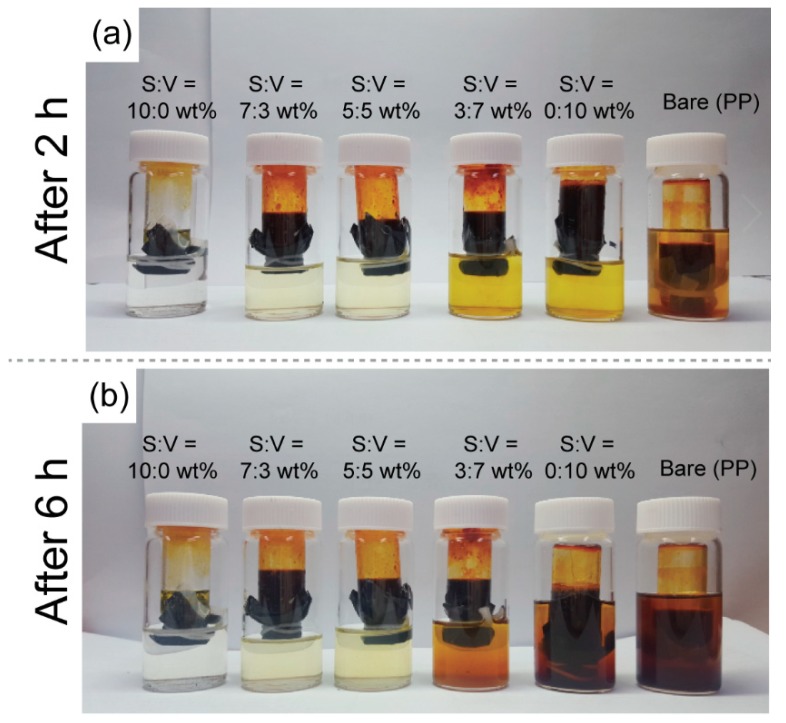
Digital camera images of polysulfide diffusion experiments after (**a**) 2 h and (**b**) 6 h. The inner glass tube filled with a mixture of polysulfide (Li_2_S_6_) and control electrolyte (DME/DOL, 1:1 by vol.) was wrapped with various types of separators, while the outer glass tube was filled with control electrolyte. S:V indicates the weight ratio of Super-P to VGCF.

**Figure 3 nanomaterials-09-00436-f003:**
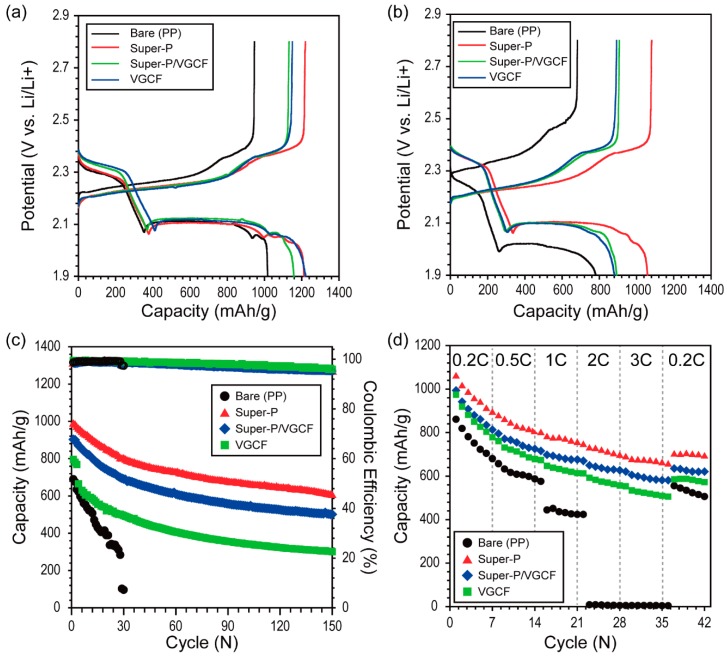
Charge/discharge potential profiles during (**a**) pre-cycling (current density = C/5) and (**b**) the first cycling for rate capability test (associated with Figure 3d); (**c**) cycle performance for the Li–S batteries containing various carbon-coated separators, respectively (current density = 1 C); and (**d**) rate cyclability (charging current density was varied from C/5 to 3 C, while the discharging current density was fixed at C/5). Super-P/VGCF consisted of Super-P:VGCF = 5:5 by wt.%.

**Figure 4 nanomaterials-09-00436-f004:**
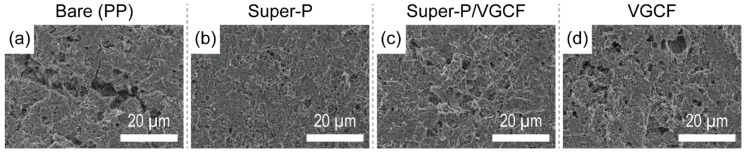
SEM images of the surfaces of (**a**)–(**d**) sulfur cathodes after 20 cycles (corresponding to Figure 3c).

**Figure 5 nanomaterials-09-00436-f005:**
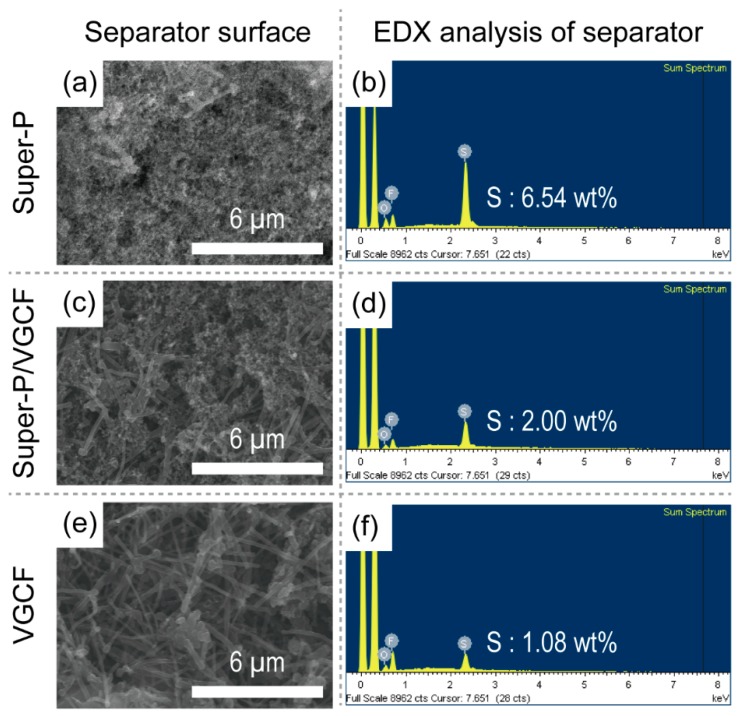
(**a**,**c**,**e**) SEM images and (**b**,**d**,**f**) EDX elemental analysis of the surfaces of the separators on the cathode side after 20 cycles (corresponding to Figure 3c).

**Figure 6 nanomaterials-09-00436-f006:**
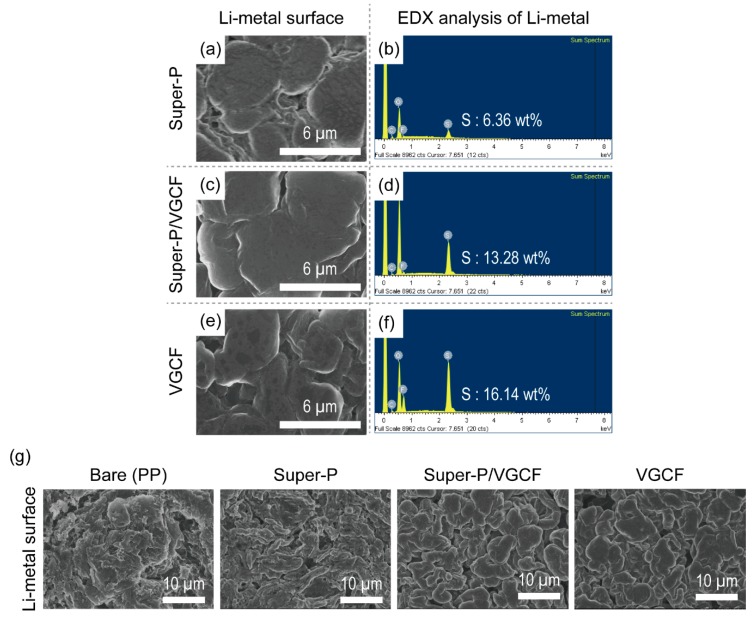
(**a**,**c**,**e**) SEM images and (**b**,**d**,**f**) EDX elemental analysis of the Li metal after 20 cycles (corresponding to Figure 3c); (**g**) SEM images of the Li metal after 50 cycles (corresponding to Figure 3c).

**Figure 7 nanomaterials-09-00436-f007:**
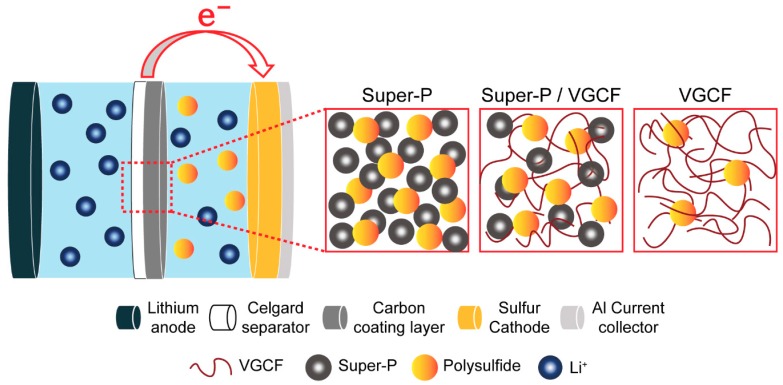
Schematic representation of polysulfide immobilization by Super-P-rich separators in Li–S cells. The black and yellow spheres represent lithium and sulfur particles, respectively.

**Table 1 nanomaterials-09-00436-t001:** Physical properties of bare polypropylene (PP) and different conductive additives (Super-P, Super-P/VGCF, VGCF).

	Celgard 2400	Super-P	Super-P/VGCF	VGCF
Thickness (µm)	25	34	35	34
Gurley number (s∙100 mL^−1^)	546.4	633.2	583.2	570.6
Surface resistance (mΩ∙cm)	N.A.	286.1	165.2	84.3

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
