# Peer review of "Effect of Varying the Ratio of Carbon Black to Vapor-Grown Carbon Fibers in the Separator on the Performance of Li–S Batteries"

_nanomaterials, 2019, doi:10.3390/nano9030436_

Round 1
Reviewer 1 Report
Authors evaluated polysulfide shuttle suppression capabilities of different carbon particles coated on polymer separator (PP) in lithium sulfur batteries (LSBs). The electrochemical performance of separators with the ratio of Super P to vapor-grown-carbon-fibers (100:0, 50:50, 0:100, w/w) was compared in terms of gravimetric capacity and rate capabilities. It was claimed that the Super P-coated separator is superior to other separators because its dense packing with low porosity inhibit the migration of soluble lithium polysulfides to lithium metal anode. Similar concept using Ketjen black coated on Celgard 2400 has already been reported somewhere else. (2016, RSCAdv, Zhao et al., 6, 13680) This work might contrast to the previous one by revealing the fact that porous structure of the coating layer has significant impact on the suppression capabilities. This might however require more elaboration to convince readers of Nanomaterials. Below are my suggestions:
1. The information in figure 2 does not match with the caption. (6h, 12h). Please correct this.
2. The description about the mechanism of charge process is incorrect. (page 6) Please correct this.
3. If C/5 is 0.27 mA/cm2, sulfur loading mass is about 0.9 mg/cm2. But, it is described that the areal loading of sulfur ranges from 1.3 mg/cm2 to 1.5 mg/cm2. Please correct this. (page 7)
4. It is important to mention specific information and evidence in SEM images of sulfur cathodes after cycling. What is formed in the surface of sulfur cathode? When was the cell disassembled, after discharge or charge? There is no explanation why the morphology of the solid discharge products changes in figure 4 after cycling.
5. The discharge profile in the GCPL test shown in fig 3(a) is unusual. Why is it unstable at ~2.0 V vs. Li/Li+?
6. It would be more informative for readers if authors can show SEM images of Li metal anodes after enough cycling, for example, 50 or 100 cycles in figure 6. They might be able to reveal the drastic change in the surface structure of lithium metal resulting from polysulfide shuttling. Please also add SEM image of LSB cell using Li metal with bare pp for clear comparison.
7. I wonder whether LSB cells with the Super P coated separator would be able to show the same result at a sulfur loading higher than 6 mg/cm2. If Li2S is produced in large quantities, it can be assumed that it will block the pores of separator with dense surface and induce a breakdown of cell. What do the authors think about this?
8. Please elaborate more on the fundamental description about Gurley number and the basis on its calculation. It is quite ambiguous to correlate Gurley number with electrochemical characteristics.
9. Please add cycle performance at lower current densities, for example, 0.3C or 0.1C, into figure 3(c). It seems to me that low C-rate conditions are tougher to suppress polysulfide dissolution than the high C-rate conditions.
10. It can be inferred that sulfur content on lithium metal in figure 6 is originated not only from discharge product, but also from LiTFSI salt. Direct evidence on the source of surface sulfur on Li metal anode must be provided.
11. Irrespective of porous structure of carbon coating layer, it could be just increased carbon content that improves the capacity. Please compare how much increase in discharge capacity or cycle stability is achieved by using total mass of cathode plus carbon coating layer.
Author Response
As can be seen in the attached Word file, all the comments raised by them were to be thoroughly answered.

Reviewer 2 Report
This article reports the carbon-coated separator for Li-S battery. Separators are very important for this battery system. Authors tried the Super-P and VGCF coat.
In Figure 7, carbon layer was shown adjacent to cathode with high surface conductivity in the case of super P.
Authors insist Gurley number and tortuosity is important, however, the difference in cycling performance of Superp+ VGCF and VGCF was larger comparing the Gurley number difference.
Porosity or density of each carbon layer was not shown in the article.
The carbon layer may be the part of the cathode and discharge-charge reaction seems take place in the pore of the carbon on the separator.
And some collection is necessary as below;
Authors use "bare electrolyte" however, "bare" is not likely to be used as this way.
"Figure 2. Digital camera images of polysulfide diffusion experiments after (a) 6 h and (b) 12 h" caption disagree the statement in the figure " (a) 2h (b) 6h".
Author Response

(The authors gave the same response as above.)

Reviewer 3 Report
In this study, the effect of varying the ratio of carbon black to vapor grown carbon fibers in the separator on the performance of Li–S batteries was investigated. This stduy would be accepted for publication after the following comments were carefully addressed.
The carbon coated separators were already proposed to improve the performance of Li-S batteries, i.e., Refs. 25 to 28. The authors should pointed out what is the difference between their concept and the reported studies, so that clarifying the motivation of this study.
The average thickness of each carbon-coating layer was about 10 um. How about the total carbon loading (mg/cm2) of each modified separator? I wonder if the carbon loading will affect the performance or not. It is suggested providing the controll experiments for better understanding.
In line 219 of P. 7, the rate capabilities should be Figure 3b.
Only using the EDX elemental analysis may not strongly support that the migration of the soluble polysulfide was efficiently inhibited by the Super-P-rich separators. Other evidences should be provided to convince the readers.
Author Response

(The authors gave the same response as above.)

Round 2
Reviewer 2 Report
Authors almost understood the problem lying in the usage of the separator as they tried and they have corrected the manuscript. There might be still some point to discuss, but this manuscript could be published as a paper in this Journal.
Author Response
Response to Reviewer #2
Authors almost understood the problem lying in the usage of the separator as they tried and they have corrected the manuscript. There might be still some point to discuss, but this manuscript could be published as a paper in this Journal.
Response: We would like to thank you for your consideration of our manuscript. The Reivewer #2 has requested additional language check for publication, but needs to postpone for a while. Now we are in the process of appealing Nanomaterials’ editorial office for another reviewer who made unreasonable decisions with the belief that a single manuscript should cover all the detailed information about the Li−S battery system. After this process, we will amend our manuscript.

Reviewer 3 Report
Most comments were not considered and adressed because authors argued they did not have enogh time. It's strong suggested the editor can give the authors enough time to address the comments.
Author Response
Response to Reviewer #3
Most comments were not considered and addressed because authors argued they did not have enough time. It's strong suggested the editor can give the authors enough time to address the comments
Response: There is an ancient Indian story about six blind folded men who are trying to understand what an elephant is. The one who just feels the trunk, thinks an elephant looks like a snake. Other people who feel the different parts think it is a fan, rope, spear, wall, tree, etc (Fig. R1). The reason everyone is telling the elephant differently because each one touched the different part of the elephant. This observation is not perfect, but is the only way to find out what an elephant is.
Figure R1. Blind men and the elephant
(Source: https://balajiviswanathan.quora.com/Lessons-from-the-Blind-men-and-the-elephant)
The lessons learned in this story apply equally to Li−S batteries, especially carbon-coated separators for Li−S battery systems. Li−S battery systems consist of Li metal anode, sulfur cathode, electrolyte, and separator, each of which plays an important role individually and/or interactively with the electrochemical performance of Li−S batteries. Unfortunately, however, each of constituent in Li−S battery systems contains complex technical elements that are not clearly understood until now. For this reason, it can be seen that if one is not a review paper, one article, even if the impact factor index is high, deals with one technical problem at a time. Practically speaking, it is impossible to include all the detailed experiments over entire issue of the system in a single manuscript.
The grounds for this claim are based on the following points:
(1) In the fourth comment of the first revision round, the Reviewer #3 asked authors to show the effect of polysulfide on Li metal anodes.
Response: It has already widely known that the capacity fade of Li−S battery system is ascribed to the high solubility of polysulfide intermediates, which react with the Li metal or is deposited elsewhere in the cell (i.e., separator) [1]. As shown in Figure R2, polysulfide forms Li2S and Li2S2 on the Li metal surface [1]. In contrast, the chemical species of −CF3, SO42−, SO32−, and RSO2R’are the discharge product from LiTFSI [2]. Sulfur element can be attributed to both polysulfide and LiTFSI. The exact electrochemical reaction on Li metal, however, is not clearly understood at present. Furthermore, the relationship between polysulfide and LiTFSI for electrochemical decomposition during discharge has not yet been clearly understood.
Figure R2. Schematic illustration of the polysulfide shuttle mechanism [1].
Taking this into account, we believe the detailed investigation on Li metal surface is far beyond the scope of the current study. We have studied many previous studies dealing with the modification of the separator and sulfur cathode for Li−S batteries, but no studies have been found on Li metal surface investigations other than EDX. Please check the previous studies listed in Table R1 below.
Table R1. A list of previous studies dealing with modification of separator or sulfur cathode for Li−S batteries.
Journal Source | Title |
ACS Energy Lett. 1 (2016) 46 | High-energy-density lithium-sulfur batteries based on blade-cast pure sulfur electrodes |
Nat. Commun. (2015) 67760 | Long-life Li/polysulphide batteries with high Sulphur loading enabled by lightweight three-dimensional nitrogen/Sulphur-codoped graphene sponge |
Adv. Mater. 27 (2015) 1694 | A facile layer-by-layer approach for high-areal-capacity sulfur cathodes |
J. Phys. Chem. Lett. 5 (2014) 3986 | Activated Li2S as a high-performance cathode for rechargeable lithium-sulfur batteries |
J. Phys. Chem. Lett. 5 (2014) 1978 | High-performance Li-S batteries with an ultra-lightweight MWCNT-coated separator |
Adv. Funct. Mater. 24 (2014) 5299 | Bifunctional separator with a light-weight carbon-coating for dynamically and statically stable lithium-sulfur batteries |
Adv. Mater. 26 (2014) 7352 | A polyethylene glycol-supported microporous carbon coating as a polysulfide trap for utilizing pure sulfur cathodes in lithium-sulfur batteries |
Nat. Commun. 3 (2012) 1166 | Lithium-sulphur batteries with a microporous carbon paper as a Bifunctional interlayer |
Chem. Commun. 48 (2012) 8817 | A new approach to improve cycle performance of rechargeable lithium-sulfur batteries by inserting a free-standing MWCNT interlayer |
(2) In the second comment, the Reviewer #3 asked authors to show the effect of carbon loading on Li−S battery performance.
Response: Threr are two main control factors for carbon-coated separators: (1) Coating thickness and (2) carbon loading. Both are importnat because they affect the porosity and polysulfide absorption capacity of the coated separators. Unfortuantely, because of the different physical properties, such as the density of Super-P and VGCF, we cannot control the two main factors equally in a single experiment. In our study, we focused on coating thickness because of the ease of control.
Remarks
Considering the backgrounds of our appeal mentioned above, however, we believe the detailed experiments performed to fulfill the Reviewer #3’s comments have values that can be published separately in different journals.
Given these backgrounds and the efforts we made so far (please see below for the reponses to reviewers for the first revision round we prepared), we sincerely ask the editorial office to have a more positive and generous view of our work.
1. Adelhelm, P.; Hartmann, P.; Bender, C.L.; Busche, M.; Eufinger, C.; Janek, J. From lithium to sodium: cell chemistry of room temperature sodium–air and sodium–sulfur batteries. Beil. J. Nanotech. 2015, 6, 1016-1055.
2. Li, W.; Yao, H.; Yan, K.; Zheng, G.; Liang, Z.; Chiang, Y.-M.; Cui, Y. The synergetic effect of lithium polysulfide and lithium nitrate to prevent lithium dendrite growth. Nat. Commun. 2015, 6, 7436.
